# CHATKBQA: A GENERATE-THEN-RETRIEVE FRAMEWORK FOR KNOWLEDGE BASE QUESTION ANSWERING WITH FINE-TUNED LARGE LANGUAGE MODELS

## ABSTRACT

Knowledge Base Question Answering (KBQA) aims to answer natural language questions over large-scale knowledge bases (KBs), which can be divided into two research components: knowledge retrieval and semantic parsing. However, three core challenges remain: inefficient knowledge retrieval, retrieval errors adversely affecting semantic parsing and the complexity of previous KBQA methods. We introduce ChatKBQA, a novel generate-then-retrieve KBQA framework built on fine-tuning open-source LLMs such as Llama-2, ChatGLM2, and Baichuan2 in the era of large language models (LLMs). ChatKBQA proposes first generating the logical form with fine-tuned LLMs, then retrieving and replacing entities and relations utilizing an unsupervised retrieval method, which improves both generation and retrieval more directly. Experiment results show that ChatKBQA achieves new state-of-the-art performance on standard KBQA datasets, WebQSP, and ComplexWebQuestions (CWQ). This work can also be regarded as a new paradigm for combining LLMs with knowledge graphs (KGs) for interpretable and knowledge-required question answering. Our code is publicly available at `https://anonymous.4open.science/r/ChatKBQA`.

## 1 INTRODUCTION

Knowledge Base Question Answering (KBQA) is a classical NLP task to answer natural language questions based on facts over a large-scale knowledge base (KB), such as Freebase (Bollacker et al., 2008), Wikidata (Vrandečić & Krötzsch, 2014), and DBpedia (Auer et al., 2007), which are composed of structured knowledge graphs (KGs) built from triples consisting of (head entity, relation, tail entity). Previous KBQA methods primarily addressed two core issues: knowledge retrieval (Yao et al., 2007) and semantic parsing (Berant et al., 2013). Knowledge retrieval mainly aims to locate the most relevant entities, relations, or triples according to the question from KB, narrowing the scope of consideration. Then, semantic parsing essentially converts the question from unstructured natural language into a structured logical form (such as S-expression (Gu et al., 2021)), which can then be converted into an executable graph database query language (such as SPARQL (Pérez et al., 2006)) to obtain precise answers and interpretable paths.

Previous KBQA work (Zhang et al., 2022a; Oguz et al., 2022; Jiang et al., 2023a) proposed different knowledge retrieval methods with technologies of named entity recognition (NER) (Devlin et al., 2019), entity linking (Li et al., 2020) or subgraph retrieval (Zhang et al., 2022a). Then, they leveraged the retrieved factual triples to directly derive answers to questions using a seq2seq model such as T5 (Raffel et al., 2020). Others (Ye et al., 2022; Hu et al., 2022b; Shu et al., 2022; Yu et al., 2023) first retrieved relevant triples, and then performed semantic parsing to generate a logical form, which can be converted to according SPARQL query to be executed over KB to fetch the answers.

Despite this, three main challenges remain, as shown on the left side of Figure 1. (1) **Low retrieval efficiency.** Traditional methods first identify the span of candidate entities and then do entity retrieval and relation retrieval. Since the structure of natural language problems differs from KB facts, most approaches require training dedicated models for extraction and linking inefficiently. (2) **Incorrect retrieval results will mislead semantic parsing.** Previous methods have utilized retrieved triples also as input of reference to the seq2seq model along with the original question. However,

Figure 1: Comparison of examples of the previous retrieve-then-generate KBQA framework (left) and our proposed generate-then-retrieve KBQA framework, ChatKBQA (right). Note that labels such as "g.g.a" etc. in the computational graph are acronyms for relation names such as "government.government_position_held.appointed_by".

since the retrieved triples are not always accurate, they adversely impact semantic parsing outcomes. Additionally, if there are numerous retrieved triples, the seq2seq model requires a much longer context length. (3) **Multiple processing steps make KBQA a redundantly complex task.** Previous work decomposed the KBQA task into multiple sub-tasks (Hu et al., 2022b; Shu et al., 2022), forming a complex pipeline, which made reproduction and migration challenging. In the era when large language models (LLMs) (OpenAI, 2023; Zhao et al., 2023; Pan et al., 2023) are restructuring traditional NLP tasks (Chung et al., 2022; Wang et al., 2023a;b), a more straightforward solution utilizing LLMs to reformulate the traditional KBQA paradigm is promising.

To overcome these challenges, we introduce **ChatKBQA**, a novel generate-then-retrieve KBQA framework based on open-source LLMs, such as Llama-2-7B (Touvron et al., 2023), ChatGLM2-6B (Zeng et al., 2023) and Baichuan2-7B (Yang et al., 2023). As illustrated on the right side of Figure 1, ChatKBQA simplifies KBQA into two efficient phases: generating logical forms and then retrieving relevant entities and relations. **In the generation phase**, leveraging instruction tuning (Ouyang et al., 2022), fine-tuned LLMs exhibit high accuracy in semantic parsing of natural language questions without retrieval. The generated logical forms are not only mostly correct in skeleton (entities and relations masked) but also semantically consistent or close to the ground truth in terms of entities and relations. **In the retrieval phase**, ChatKBQA proposes an unsupervised retrieval method that employs phrase-level semantic retrieval within knowledge bases to improve generation accuracy and retrieval efficiency further. Additionally, ChatKBQA features a plug-and-play characteristic, ensuring compatibility with various LLMs and semantic retrieval models, making it a flexible, high-performance solution for KBQA tasks.

To test the performance of our proposed framework, we conduct experiments on two standard KBQA datasets, WebQSP (Yih et al., 2016b) and ComplexWebQuestions (CWQ) (Talmor & Berant, 2018), with both settings of using and not using golden entities. The experimental results demonstrate that ChatKBQA achieves a new state-of-the-art performance in the KBQA task. We also set up additional experiments to validate that our generate-then-retrieve approach improves both generation and retrieval results efficiency. Finally, we also discuss how insights from this framework lead us to envision future combinations of LLMs and KGs.

## 2 RELATED WORK

**Knowledge Base Question Answering.** Existing Knowledge Base Question Answering (KBQA) methods can be broadly categorized into Information Retrieval-based (IR-based) and Semantic Parsing-based (SP-based) methods. (1) IR-based methods (Miller et al., 2016; Sun et al., 2018; 2019; Saxena et al., 2020; He et al., 2021; Shi et al., 2021; Yan et al., 2021; Zhang et al., 2022b; Oguz et al., 2022; Dong et al., 2023) primarily retrieve relevant factual triples or text from Knowledge Bases (KBs) based on natural language questions, forming a subgraph to determine answers. (2)

On the other hand, SP-based methods focus on translating questions into logical forms executable against KBs, such as SPARQL, query graph, and S-expression. Some SP-based approaches (Yih et al., 2016a; Chen et al., 2019; Lan et al., 2019; Bhutani et al., 2019; Lan & Jiang, 2020; Sun et al., 2020; Gu et al., 2021; Sen et al., 2021; Jiang et al., 2023b; Atif et al., 2023; Chen et al., 2023; Jiang et al., 2023a; Gu et al., 2023; Sun et al., 2023) utilize strategies of step-wise query graph generation and search for semantic parsing. Alternatively, other SP-based methods (Chen et al., 2021; Das et al., 2021; Ye et al., 2022; Cao et al., 2022; Shu et al., 2022; Gu & Su, 2022; Hu et al., 2022b; Liu et al., 2022b; Xie et al., 2022; Yu et al., 2023; Zhang et al., 2023) employ sequence-to-sequence models to generate S-expressions completely and offer various enhancements to the semantic parsing process. In this paper, our proposed ChatKBQA is the first SP-based KBQA method using fine-tuned LLMs, which innovatively proposes a generate-then-retrieve approach to simplify KBQA method.

**Large Language Models.** With the launch of ChatGPT and GPT-4 (OpenAI, 2023), displaying the prowess of decoder-only large language models (LLMs) with a vast number of parameters that exhibit emergent phenomena, many traditional NLP tasks are becoming simplified (Zhao et al., 2023). Subsequently, open-source LLMs like Llama-2-7B (Touvron et al., 2023), ChatGLM2-6B (Zeng et al., 2023) and Baichuan2-7B (Yang et al., 2023) emerged and can be supervisedly fine-tuned (SFT) using Parameter-Efficient Fine-Tuning (PEFT) technologies (Mangrulkar et al., 2022) such as LoRA (Hu et al., 2022a), QLoRA (Dettmers et al., 2023), P-Tuning v2 (Liu et al., 2022a), and Freeze (Geva et al., 2021), enhancing the capabilities of LLMs for specific tasks.

**Knowledge Retrieval for KBQA.** General retrieval methods are typically divided into traditional lexical models, such as BM25 (Robertson & Zaragoza, 2009), and dense retrieval models, such as Dense Passage Retrieval (DPR) (Karpukhin et al., 2020), SimCSE (Gao et al., 2021), and Contriever (Izacard et al., 2022). In the KBQA task, to better utilize knowledge related to the question from knowledge bases, efficient retrieval algorithms are needed to fetch the most relevant knowledge. ELQ (Li et al., 2020) and FACC1 (Evgeniy et al., 2013) are commonly used entity retrieval methods. In this paper, our KBQA framework, ChatKBQA, retrieves corresponding entities and relations after generating the logical form by FACC1 and SimCSE, respectively, in an unsupervised manner to improve the efficiency of retrieval.

## 3 PRELIMINARIES

**Definition 1: Knowledge Base (KB).** A KB $\mathcal{K} = \{(s,r,o)|s \in \mathcal{E}, r \in \mathcal{R}, o \in \mathcal{E} \cup \mathcal{L}\}$ is an RDF graph consisting of the triples $(s,r,o)$ where $s$ is an entity, $r$ is a relation , and $o$ can be an entity or a literal. Each entity $e \in \mathcal{E}$ in the entity set $\mathcal{E}$ is represented by a unique ID, e.g., `e.id="m.0fm2h"`, and can be queried to get the English label of the entity as `e.label="Benjamin Netanyahu"`. Each relation $r \in \mathcal{R}$ in the set of relations $\mathcal{R}$ consists of multiple levels of labels, e.g. `r="government.government_position_held.appointed_by"`. Besides, a literal $l \in \mathcal{L}$ is usually "integer" (e.g., `l="32"`), "float" (e.g., `l="3.2"`), "year" (e.g., `l="1999"`), "year&month" (e.g., `l="1999-12"`), or "date" (e.g., `l="1999-12-31"`).

**Definition 2: Logical Form.** A logical form is a structured representation of a natural language problem. Taking the S-expression as an example, a logical form usually consists of projection and various operators. Projection operation represents a one-hop query of a triple $(s,r,o)$ on $s$ or $o$, where, $(?,r,o)$ is denoted as `(JOIN r o)`, while $(s,r,?)$ is denoted as `(JOIN (R r) s)`. Various operators include "AND" `(AND E1 E2)` to denote taking the intersection of $E_1$ and $E_2$, "COUNT" `(COUNT E1)` to denote counting $E_1$, "ARGMAX" `(ARGMAX E1 r)` to denote taking the max literal obtained after the projection of $E_1$ in the $r$ relation, "ARGMIN" `(ARGMIN E1 r)` to denote taking the min literal obtained after the projection of the $r$ relation for $E_1$, "GT" `(GT E1 l)` means to take the portion of $E_1$ that is greater than $l$, "GE" `(GE E1 l)` to denote taking the part of $E_1$ greater than or equal to $l$, "LT" `(LT E1 l)` to denote taking the part of $E_1$ less than $l$, "LE" `(LE E1 l)` to denote taking the part of $E_1$ which is less than or equal to $l$, where $E_1$ or $E_2$ denote a sublayer logical form.

**Problem Statement.** For KBQA task, given a natural language question $Q$, and a knowledge base $\mathcal{K}$, we need to first convert $Q$ into a logical form $F = \texttt{Sp}(Q)$, where $\texttt{Sp}(.)$ is a semantic parsing function. Then convert $F$ to the equivalent SPARQL query $q = \texttt{Convert}(F)$, where $\texttt{Convert}(.)$ is the fixed conversion function. Finally the final set of answers $A = \texttt{Execute}(q|\mathcal{K})$ is obtained by executing $q$ against $\mathcal{K}$, where $\texttt{Execute}(.)$ is the query execution function.

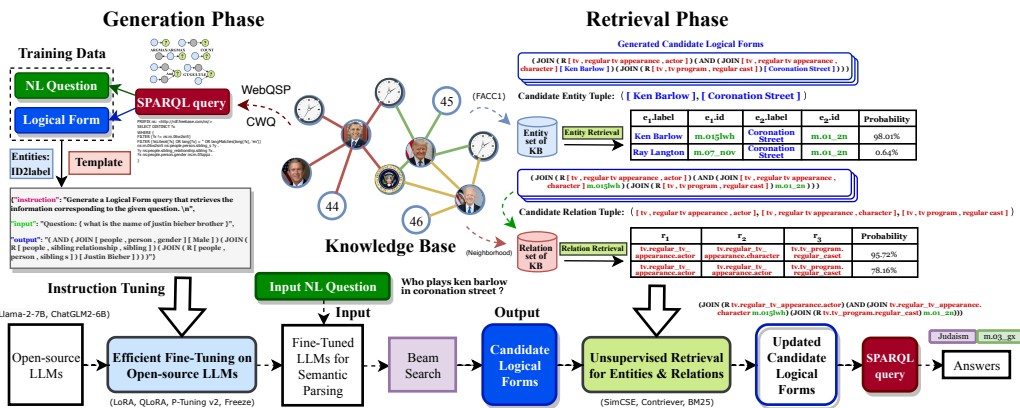

Figure 2: The overview of ChatKBQA framework for generate-then-retrieve KBQA method with fine-tuned LLMs and unsupervised retrieval for entities and relations in candidate logical forms.

# 4 METHODOLOGY

ChatKBQA is a generate-then-retrieve KBQA framework with fine-tuned LLMs as illustrated in Figure 2. First, the ChatKBQA framework needs to efficiently fine-tune an open-source LLM based on the (natural language question, logical form) pairs in the KBQA dataset by instruction tuning. The fine-tuned LLM is then used to convert the new natural language questions to according candidate logical forms by semantic parsing. Then, ChatKBQA retrieves the entities and relations in these logical forms at the phrase level, and searches for the logical forms that can be executed against KB after being converted to SPARQL. Finally, the converted SPARQL is used to generate the final set of answers, resulting in interpretable and knowledge-required responses to natural language questions.

## 4.1 EFFICIENT FINE-TUNING ON LLMS

To construct the instruction fine-tuning training data, ChatKBQA first converts the SPARQL corresponding to the natural language questions of the train set in the KBQA dataset into equivalent logical forms, and then replaces the entity IDs (e.g., "`m.06w2sn5`") in these logical forms with the corresponding entity tags (e.g., "`[ Justin Bieber ]`"), to let LLMs understand entity labels better than meaningless entity IDs. We then combine the natural language question (e.g. "`What is the name of justin bieber brother?`") and the processed corresponding logical form (e.g. "`(AND (JOIN [ people , person , gender ] [ Male ]) (JOIN (R [ people , sibling relationship , sibling ]) (JOIN (R [ people , person , sibling s ]) [ Justin Bieber ])))`") as "input" and "output" respectively, and add "instruction" as "`Generate a Logical Form query that retrieves the information corresponding to the given question.`" constitutes the instruction fine-tuning training data for open source LLMs.

ChatKBQA employs Parameter Efficient Fine-Tuning (PEFT) (Mangrulkar et al., 2022) techniques including various efficient fine-tuning methods, such as LoRA (Hu et al., 2022a), QLoRA (Dettmers et al., 2023), P-tuning v2 (Liu et al., 2022a), and Freeze (Geva et al., 2021), to minimize the cost of fine-tuning LLMs with a large number of parameters. ChatKBQA can switch between all the above fine-tuning methods as well as open-source LLMs, such as Llama-2-7B (Touvron et al., 2023), ChatGLM2-6B (Zeng et al., 2023) and Baichuan2-7B (Yang et al., 2023).

## 4.2 LOGICAL FORM GENERATION BY FINE-TUNED LLMS

Through fine-tuning, the LLMs have acquired expertise in semantic parsing, enabling them to convert natural language questions into logical forms. We apply the fine-tuned LLMs to perform semantic parsing on the new questions in the test set, and observe that approximately 63% of the samples match the ground truth logical forms exactly. When employing beam search, the set of candidate logical forms $\mathcal{C}$ generated by our LLMs includes approximately 74% of the instances with

correct logical forms, indicating that fine-tuned LLMs possess effective learning and parsing abilities for semantic parsing tasks. In addition, by replacing the entities and relations in the candidate logical forms with "[]" (for example, "(AND (JOIN [] []) (JOIN (R []) (JOIN (R []) [])))"), more than 91% of the samples contain the candidate skeleton. Hence, the next step involves retrieving the entities and relations in the logical form with the corresponding ones from the KB to enhance performance further.

## 4.3 Unsupervised Retrieval for Entities and Relations

Due to the strong generative capabilities of fine-tuned LLMs for logical form skeletons, we employ an unsupervised retrieval approach during the retrieval phase. This method involves subjecting the entities and relations in the candidate logical forms to phrase-level semantic retrieval and replacement. The result is a final logical form that can be executed as a SPARQL query against the KB.

Specifically, as shown in the Algorithm 1, the input is the generated candidate logical form list $\mathcal{C}$, and we traverse each of these logical forms $F$ in order. First, we perform the entity retrieval. For each entity $e$ in $F$, we compute the similarity $s_e \leftarrow \text{SimiEntities}(e, e')$ with the label of each entity $e'$ in the knowledge base $\mathcal{K}$ entity set $\mathcal{E}$. We sort the retrieved entities based on the similarities, take the top $k_e$ and greater than the threshold $t_e$ to get the retrieval result for that entity $e_{list} \leftarrow \text{TopKwithThreshold}(e_{list}, k_e, t_e)$. Function $\text{PermuteByEntity}$ performs permutation on the retrieved entities at each position, and we get the result $F_{list}$ after entity retrieval. Based on probabilities in $F_{list}$, we take top $k_1$ and greater than threshold $t_1$ to get a new candidate logical form list $\mathcal{C}'.append(\text{TopKwithThreshold}(F_{list}, k_1, t_1))$.

Then, we perform the relation retrieval. Similar to entity retrieval, but different in that for each relation $r$ in $F \in \mathcal{C}'$, we compute the similarity $s_r \leftarrow \text{SimiRelations}(r, r')$ with each candidate relation $r'$ according to the neighborhood of entity set of the logical form $\mathcal{E}_F$. We also sort the retrieved relations according to the similarities, take the top $k_r$ and greater than the threshold $t_r$ to get the retrieval result $r_{list} \leftarrow \text{TopKwithThreshold}(r_{list}, k_r, t_r)$. By permuting the retrieval results of the relations at each position, we get the result $F_{list}$ after relation retrieval and then take top $k_2$ and greater than the threshold $t_2$ to get a new list of candidate logical forms $\mathcal{C}''.append(\text{TopKwithThreshold}(F_{list}, k_2, t_2))$.

---

**Algorithm 1:** Unsupervised Retrieval

**Input** : Candidate logical form list generated from LLM $\mathcal{C}$, top-$k$ threshold $k_e, k_r, k_1, k_2$, probability threshold $t_e, t_r, t_1, t_2$, the entity set of knowledge base $\mathcal{E}$

**Output:** The equivalent SPARQL query $q$

$\mathcal{C}' \leftarrow \emptyset$;
**foreach** $F \in \mathcal{C}$ **do**
  **foreach** $e \in F$ **do**
    $e_{list} \leftarrow \emptyset$;
    **foreach** $e' \in \mathcal{E}$ **do**
      $s_e \leftarrow \text{SimiEntities}(e, e')$;
      $e_{list}.append((e', s_e))$;
    $e_{list} \leftarrow \text{TopKwithThreshold}(e_{list}, k_e, t_e)$;
    $F.attach(e_{list})$;
  $F_{list} \leftarrow \text{PermuteByEntity}(F)$;
  $\mathcal{C}'.append(\text{TopKwithThreshold}(F_{list}, k_1, t_1))$;

$\mathcal{C}'' \leftarrow \emptyset$;
**foreach** $F \in \mathcal{C}'$ **do**
  **foreach** $e \in F$ **do**
    $r_{list} \leftarrow \emptyset$;
    **foreach** $r \in \text{Neighborhood}(\mathcal{E}_F)$ **do**
      $s_r \leftarrow \text{SimiRelations}(r, r')$;
      $r_{list}.append((r', s_r))$;
    $r_{list} \leftarrow \text{TopKwithThreshold}(r_{list}, k_r, t_r)$;
    $F.attach(r_{list})$;
  $F_{list} \leftarrow \text{PermuteByRelation}(F)$;
  $\mathcal{C}''.append(\text{TopKwithThreshold}(F_{list}, k_2, t_2))$;

**foreach** $q \in \mathcal{C}''$ **do**
  $q = \text{Convert}(F)$;
  **if** $q$ *is valid to execute* **then**
    **return** $q$;

**return** $\emptyset$;

---

Given a query, unsupervised retrieval methods such as SimCSE (Gao et al., 2021), Contriever (Izacard et al., 2022), and BM25 (Robertson & Zaragoza, 2009), require no additional training to identify the top k most semantically similar candidates from the set of retrieved answers. ChatKBQA can switch between all the above unsupervised retrieval methods for entity retrieval and relation retrieval.

## 4.4 Interpretable Query Execution

After retrieval, we get a final candidate logical form list $\mathcal{C}''$, which we sequentially iterate through the logical form $F \in \mathcal{C}''$ and convert to the equivalent of the SPARQL query $q = \text{Convert}(F)$. When the first $q$ that can be executed against KB $\mathcal{K}$ is found, we execute to get the final answer set $A = \text{Execute}(q|\mathcal{K})$. With this approach, we can also get a complete reasoning path for natural language questions based on SPARQL query with good interpretability. To summarize, ChatKBQA proposes a thought taking both the advantages of using LLMs to do natural language semantic parsing for graph query generation and calling external KBs to interpretably reason with queries.

## 5 EXPERIMENTS

This section presents the experimental setup, results, and analysis. We answer the following research questions (RQs): **RQ1**: Does ChatKBQA outperform other KBQA methods? **RQ2**: Does the main components of ChatKBQA work? **RQ3**: Why use Generate-then-Retrieve method instead of Retrieve-then-Generate method? **RQ4**: Why use fine-tuned open-source LLMs instead of calling ChatGPT or training traditional T5 models? **RQ5**: Does Generate-then-Retrieve method improve the efficiency of retrieval? **RQ6**: Is ChatKBQA plug-and-play? **RQ7**: How about error analysis?

### 5.1 EXPERIMENTAL SETUP

**Datasets.** All experiments are conducted on two standard KBQA datasets: WebQuestionsSP (WebQSP) (Yih et al., 2016b) containing 4,737 natural language questions with SPARQL queries and ComplexWebQuestions (CWQ) (Talmor & Berant, 2018) containing 34,689 natural language questions with SPARQL queries. Both datasets are based on Freebase (Bollacker et al., 2008) KB.

**Baselines.** We compare ChatKBQA with numerous KBQA baseline methods, including KV-Mem (Miller et al., 2016), STAGG (Yih et al., 2016a), GRAFT-Net (Sun et al., 2018), UHop (Chen et al., 2019), Topic Units (Lan et al., 2019), TextRay (Bhutani et al., 2019) and all other KBQA methods in Section 2.

**Evaluation Metrics.** Following previous work (Hu et al., 2022b; Shu et al., 2022; Yu et al., 2023), we use $F_1$ score, Hits@1, and Accuracy (Acc) to denote coverage of all the answers, single top-ranked answer, and strict exact-match accuracy, respectively.

**Hyperparameters and Enviroment.** We fine-tune LLMs 100 epochs on WebQSP and 10 epochs on CWQ with batch size 4 and learning rate 5e-5. All experiments were done on a single NVIDIA A40 GPU (48GB), with results averaged from five randomly seeded experiments.

### 5.2 MAIN RESULT (RQ1)

For the KBQA task, Table 1 lists the experimental results for our proposed generate-then-retrieve ChatKBQA framework, with the best setup of LoRA (Hu et al., 2022a) fine-tuning Llama-2-7B (Touvron et al., 2023) (beam size = 15) on WebQSP, Llama-2-13B (Touvron et al., 2023) (beam size = 8) on CWQ, with SimCSE (Gao et al., 2021) for unsupervised retrieval, and other baseline models. We can see that ChatKBQA has a significant improvement over all existing KBQA methods on both WebQSP and CWQ datasets. The $F_1$ score,

| Model | WebQSP | | | CWQ | | |
|---|---|---|---|---|---|---|
| | F1 | Hits@1 | Acc | F1 | Hits@1 | Acc |
| KV-Mem | 34.5 | 46.7 | - | 15.7 | 21.1 | - |
| STAGG | 71.7 | - | 63.9 | - | - | - |
| GRAFT-Net | 62.8 | 67.8 | - | 32.7 | 36.8 | - |
| UHop | 68.5 | - | - | 29.8 | - | - |
| Topic Units | 67.9 | 68.2 | - | 36.5 | 39.3 | - |
| TextRay | 60.3 | 72.2 | - | 33.9 | 40.8 | - |
| PullNet | - | 68.1 | - | - | 47.2 | - |
| QGG | 74.0 | 73.0 | - | 40.4 | 44.1 | - |
| EmbedKGQA* | - | 66.6 | - | - | 44.7 | - |
| EmQL* | - | 75.5 | - | - | - | - |
| NSM+h* | 67.4 | 74.3 | - | 44.0 | 48.8 | - |
| GrailQA Ranking* | 70.0 | - | - | - | - | - |
| ReTraCk* | 74.7 | 74.6 | - | - | - | - |
| TransferNet | - | 71.4 | - | - | 48.6 | - |
| Relation Learning | 64.5 | 72.9 | - | - | - | - |
| Rigel* | - | 73.3 | - | - | 48.7 | - |
| CBR-KBQA | 72.8 | - | 69.9 | 70.0 | 70.4 | 67.1 |
| Subgraph Retrieval* | 64.1 | 69.5 | - | 47.1 | 50.2 | - |
| RnG-KBQA | 75.6 | - | 71.1 | - | - | - |
| Program Transfer* | 76.5 | 74.6 | - | 58.7 | 58.1 | - |
| TIARA* | 78.9 | 75.2 | - | - | - | - |
| UniK-QA | 79.1 | - | - | - | - | - |
| ArcaneQA | 75.6 | - | - | - | - | - |
| GMT-KBQA | 76.6 | - | 73.1 | 77.0 | - | 72.2 |
| Uni-Parser* | 75.8 | - | 71.4 | - | - | - |
| UnifiedSKG | 73.9 | - | - | 68.8 | - | - |
| UniKGQA* | 72.2 | 77.2 | - | 49.4 | 51.2 | - |
| DECAF | 78.8 | 82.1 | - | - | 70.4 | - |
| BeamQA* | - | 73.4 | - | - | - | - |
| HGNet* | 76.6 | 76.9 | 70.7 | 68.5 | 68.9 | 57.8 |
| SKP | - | 79.6 | - | - | - | - |
| StructGPT* | 72.6 | - | - | - | - | - |
| FC-KBQA | 76.9 | - | - | 56.4 | - | - |
| PanGu | 79.6 | - | - | - | - | - |
| ToG* | - | 82.6 | - | - | 69.5 | - |
| ChatKBQA (ours) | 79.8 | 83.2 | 73.8 | 77.8 | 82.7 | 73.3 |
| ChatKBQA* (ours) | **83.5** | **86.4** | **77.8** | **81.3** | **86.0** | **76.8** |

Table 1: KBQA result comparison of ChatKBQA with other baselines on WebQSP and CWQ datasets. * denotes using oracle entity linking annotations. The results of the models are mainly taken from their original paper. For our proposed ChatKBQA framework, we display the results of the best setup on WebQSP and CWQ, respectively. The best results in each metric are in **bold**.

Hits@1, and Acc are improved by about 4, 4, and 4 percentage points on WebQSP and about 4, 16, and 4 percentage points on CWQ, respectively, compared to the previous best results, which reflects ChatKBQA's superior KBQA capability to reach the new state-of-the-art performance.

## 5.3 Ablation Study (RQ2)

In order to validate the effectiveness of the generation and retrieval phases of ChatKBQA, we ablate the two phases separately. For the generation phase, we use 20%, 40%, 60%, and 80% of the training data for fine-tuning versus full training set fine-tuning. For the retrieval phase, in order to validate entity retrieval (ER) and relation retrieval (RR) separately, we removed ER or RR from the framework and obtained three simplified variants (**ChatKBQA w/o ER**, **ChatKBQA w/o RR** and **ChatKBQA w/o ER,RR**) at four different beam sizes for comparison.

**Effectiveness of LLM's Fine-tuning.** As shown in Figure 3, the performance of KBQA gets better as the training volume increases, proving the effectiveness of fine-tuning. We also observe that the F1 score has exceed 70% when only using 20% training data to fine-tune, which indicates that the fine-tuned LLMs are also effective at learning from a limited dataset.

**Effectiveness of Beam Search.** Beam search is a heuristic algorithm usually used in sequence generation tasks, which expands the search space by generating multiple highly probable logical forms instead of only one. As shown in Figure 4, an increase in beam size enhances the likelihood of executing SPARQL queries based on candidate logical forms, improving the KBQA performance.

**Effectiveness of Entity Retrieval (ER).** As shown in Figure 4, ER improves about 15 percentage points on average over no oracle entity linking in the F1 score at different beam sizes. This is because, after LLM's Fine-tuning, the generated logical forms contain entities unseen in the train set, which can be further aligned to KB after retrieving the entities from the KB entity set.

**Effectiveness of Relation Retrieval (RR).** As shown in Figure 4, RR enhances F1 score by an average of 5% across various beam sizes in ablation experiments. Although relations are rarely directly present in natural language problems, the number of thousand-level relations in the KB is still small compared to the tens of millions of entities, and the LLM perceives relational information well during fine-tuning. Thus, RR does not improve performance as much as ER, but combined with ER, RR makes KBQA perform at its best.

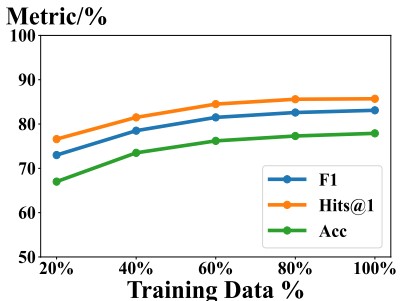

Figure 3: Ablation study in ChatKBQA generation phase to verify the effectiveness of LLM's Fine-tuning.

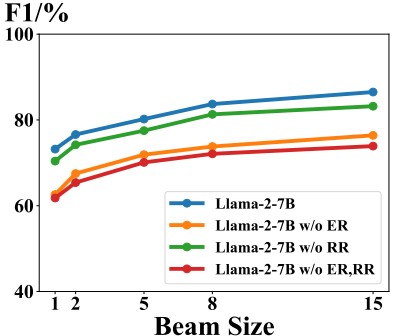

Figure 4: Ablation study in ChatKBQA retrieval phase to verify the effectiveness of Beam Search, Entity Retrieval (ER), and Relation Retrieval (RR), respectively.

## 5.4 Generate-then-Retrieve Or Retrieve-then-Generate (RQ3)

In order to verify that our proposed LLM-based Generate-then-Retrieve method is better than previous Retrieve-then-Generate methods, we add Top1, Top2, Top5, and Top10 retrieval knowledge fragments obtained in DECAF (Yu et al., 2023) to the instruction, respectively, compared with the fine-tuning of Llama-2-7B without retrieval.

| Fine-tuning Settings | WebQSP | | |
| --- | --- | --- | --- |
| | Max Token↓ | EM↑ % | BM↑ % | SM↑ % |
| Llama-2-7B w/o R | **512** | **63.5** | **74.7** | **91.1** |
| Llama-2-7B w Top1 R | 612 | 58.5 | 72.3 | 88.4 |
| Llama-2-7B w Top2 R | 712 | 59.7 | 73.6 | 89.0 |
| Llama-2-7B w Top5 R | 1012 | 55.6 | 68.3 | 85.3 |
| Llama-2-7B w Top10 R | 2012 | 53.1 | 67.9 | 84.8 |

Table 2: Comparison of whether or not utilizing retrieval results before fine-tuning Llama-2-7B for logical form generation in ChatKBQA.

As shown in Table 2, we find that without retrieval is better than with retrieval in the logical form generation in terms of extract match ratio (EM), match after beam search ratio (BM), and skeleton match ratio (SM), due to the fact that the information obtained from retrieval will **have erroneous interfering information** and **increase Max Token of instruction**, which leads to catastrophic forgetting of the original problem for LLMs

and increases the difficulty of training. At the same time, we observe that Llama-2-7B fine-tuning without retrieval achieves a BM of 74.7% and SM hits 91.1%, with good performance because of LLM's well-learned schema of entities and relations, which provides the basis for the retrieval after generation.

## 5.5 COMPARISON WITH CHATGPT AND T5 IN GENERATION PHASE (RQ4)

To illustrate why ChatKBQA chooses to fine-tune open-source generative LLMs such as Llama-2-7B and ChatGLM2-6B, we replace the LLMs in the generation phase with ChatGPT and GPT-4 (OpenAI, 2023) with API call in a zero-shot setting, T5 (Raffel et al., 2020) and Flan-T5 (Chung et al., 2022) with sequence-to-sequence training, respectively, and observe their results in Extract Match (EM) and Skeleton Match (SM) results without beam search.

**Comparison with zero-shot ChatGPT & GPT-4.** As shown in Figure 5, ChatGPT and GPT-4, although having large parametric quantities, cannot generate standard logical forms well because they aren't open-source to be fine-tuned. They can generate the SPARQL language, but it is challenging to build the correct query skeleton, entities, and relations because they cannot perceive the complex structure of the external KB well through designing prompts in limited context length.

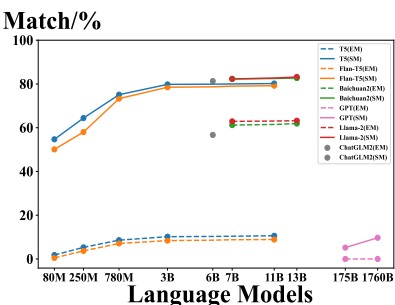

Figure 5: Comparison with other LMs in ChatKBQA generation phase.

**Comparison with fine-tuned T5 & Flan-T5.** While T5 and Flan-T5 can capture the skeletons well after fine-tuning, but the EM is only about 10%, which is much worse than the 63% of Llama-2-7B, and therefore does not guarantee subsequent unsupervised entity and relation retrieval. Fine-tuned open-source LLMs such as Llama-2-7B (Touvron et al., 2023) and ChatGLM2-6B (Zeng et al., 2023) show stronger semantic parsing ability than models such as T5 and ChatGPT, and can generate higher-quality logical forms in both EM & SM.

## 5.6 ANALYSIS OF EFFICIENCY OF RETRIEVAL IN RETRIEVAL PHASE (RQ5)

To embody the Generate-then-Retrieve method improving the efficiency of retrieval, we compare entity retrieval (ER) and retrieval (RR) after logical form generation (AG-R) with traditional retrieval from natural language questions (NL-R). We define the efficiency of retrieval as the average similarity ranging [0,1] between the text to be retrieved and the set of retrieved answers, which is scored by different retrieval models. Note that BM25 needs to be scored and then mapped to the similarity range of [0,1] by the mapping function.

**Efficiency gains in both ER & RR.** As Figure 6 shows, all three retrieval methods SimCSE (Gao et al., 2021), Contriever (Izacard et al., 2022), and BM25 (Robertson & Zaragoza, 2009) consider AG-R to be more efficient than NL-R for both ER and RR. This is due to the fact that NL-R still needs to determine the boundaries of the entities or relations. However, this step has been completed in AG-R after LLM generates the logical forms.

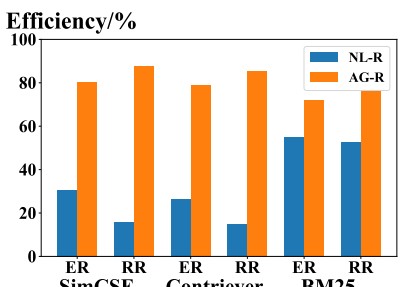

Figure 6: Comparison of retreival efficiency between retrieval from nature language questions (NL-R) and generated logical forms (AG-R) in ChatKBQA retrieval phase.

**RR has more significant efficiency gains than ER.** Moreover, although the generated logical form has fewer kinds of relations than entities in general, the relations generally exist implicitly in natural language questions. Thus, relations are more difficult to determine the boundaries than entities in natural language problems, and the generation of logical forms with the help of fine-tuned LLMs can help us to better determine the boundaries of relations, resulting in a more significant improvement in the efficiency of RR over ER.

## 5.7 Plug-and-Play Characteristics (RQ6)

ChatKBQA is a KBQA framework based on LLMs with plug-and-play characteristics that can flexibly replace three parts: LLM, efficient tuning method, and unsupervised retrieval method. We choose Llama-2-13B (Touvron et al., 2023) for LLM, LoRA (Hu et al., 2022a) for the tuning method, and SimCSE (Gao et al., 2021) for the retrieval method as the basic variant, setting the beam size for all variants to 8 for comparison. We replace Baichuan2-7B (Yang et al., 2023), Baichuan2-13B (Yang et al., 2023), ChatGLM2-6B (Zeng et al., 2023), Llama-2-7B (Touvron et al., 2023) **in the LLM part**, QLoRA (Dettmers et al., 2023), P-Tuning v2 (Liu et al., 2022a), Freeze (Geva et al., 2021) **in the tuning part**, and Con-

| ChatKBQA Framework | | | WebQSP | | |
|---|---|---|---|---|---|
| **LLMs** | **Tuning** | **Retrieval** | **F1** | **Hits@1** | **Acc** |
| Baichuan2-7B | LoRA | SimCSE | 79.1 | 81.5 | 74.1 |
| Baichuan2-13B | LoRA | SimCSE | 79.4 | 82.1 | 74.4 |
| ChatGLM2-6B | LoRA | SimCSE | 79.8 | 82.7 | 74.5 |
| Llama-2-7B | LoRA | SimCSE | 80.0 | 82.4 | 75.2 |
| **Llama-2-13B** | **LoRA** | **SimCSE** | **82.6** | **85.2** | **77.5** |
| Llama-2-13B | QLoRA | SimCSE | 81.9 | 85.0 | 76.9 |
| ChatGLM2-6B | P-Tuning v2 | SimCSE | 74.6 | 77.8 | 70.6 |
| Llama-2-13B | Freeze | SimCSE | 81.7 | 84.7 | 76.8 |
| Llama-2-13B | LoRA | Contriever | 81.5 | 83.6 | 76.8 |
| Llama-2-13B | LoRA | BM25 | 79.8 | 80.5 | 72.7 |

Table 3: Plug-and-play performance comparison of ChatKBQA framework for replacing LLMs, tuning methods, and unsupervised retrieval methods, respectively, with the beam size all set as 8.

triever (Izacard et al., 2022), BM25 (Robertson & Zaragoza, 2009) **in the retrieval part**. Benefited from the plug-and-play characteristics of ChatKBQA, as the LLMs and the methods of tuning and retrieval are upgraded, the KBQA task will be solved better with good flexibility and extensibility.

## 5.8 Error Analysis (RQ7)

We analyz the questions in the WebQSP test set that were not answered correctly by ChatKBQA without oracle entity linking, and errors can be summarized as follows.

**Logical form skeleton error** (40.10%). We discover that the majority of the errors are caused by ChatKBQA failing to provide the correct logical form skeleton for the problem, e.g. predicting "`(JOIN (R []) (JOIN (R []) []))`" as "`(JOIN (R []) [])`". This is due to the limited representation of certain complex skeletons in the train set.

**Entity retrieval error** (27.17%). Then, a portion of the samples that predicted the correct logical form skeletons, but did not retrieve the correct entities, e.g. predicting "`(JOIN (R []) m.0d3k14)`" as "`(JOIN (R []) m.07618sw)`".

**Relation retrieval error** (19.48%). In the case of successful skeleton prediction and entity retrieval, errors in relation retrieval can also lead to failed logical form generation that does not match the ground truth, e.g. predicting "`(JOIN (R finance.currency.countries_used) m.0kz1h)`" as "`(JOIN (R finance.currency.currency_code) m.0kz1h)`".

**SPARQL convertion error** (13.26%). Finally, a small proportion of the remaining errors arise from the fact that, although the generated logical form is consistent with the ground truth, it fails to execute or the answers are inconsistent when converted to SPARQL, which may be caused by the loss of the conversion from logical form to SPARQL or possibly changes in the KB.

## 6 Conclusion

In this work, we present ChatKBQA, a generate-then-retrieve framework for Knowledge Base Question Answering (KBQA) that leverages the power of modern fine-tuned large language models (LLMs). By focusing on the generation of logical forms prior to retrieval, our method offers a significant shift from traditional approaches, addressing inherent challenges such as retrieval inefficiencies and the misleading influence of retrieval errors in semantic parsing with fine-tuned open-source LLMs and unsupervised retrieval methods. Our experimental results are based on two standard KBQA benchmarks, WebQSP and CWQ, confirming that ChatKBQA achieves a new state-of-the-art performance in the KBQA domain. Moreover, the simplicity and flexibility of our framework, especially its plug-and-play characteristics, make it a promising direction for integrating LLMs with KBs for more interpretative and knowledge-required question-answering tasks.

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

APPENDIX

# A  BASELINE KBQA METHODS

In the main experiment, we compared ChatKBQA with all KBQA models in Section 2 as follows in order of publication.

**KV-Mem** (Miller et al., 2016) uses a key-value structured memory model to enhance document comprehension and question-answering by encoding facts and reasoning over them for accurate predictions.

**STAGG** (Yih et al., 2016a) presents a KBQA method using semantic parse labeling, showing improvements in query accuracy compared to relying solely on question-answer pairs.

**GRAFT-Net** (Sun et al., 2018) introduces a novel graph convolution-based neural network that enhances open domain question answering by combining information from knowledge bases and text documents into a single model.

**UHop** (Chen et al., 2019) introduces a framework for unrestricted-hop relation extraction to handle queries requiring any number of relational hops in a knowledge graph, improving the capability to answer complex and indirect questions.

**Topic Units** (Lan et al., 2019) utilizes a wide range of knowledge base units for question answering, employing a generation-and-scoring approach and reinforcement learning to enhance the identification and ranking of relevant topic units.

**TextRay** (Bhutani et al., 2019) decomposes complex questions into simpler queries, processes them individually, and combines the results, using a semantic matching model.

**PullNet** (Sun et al., 2019) presents a method that iteratively constructs a question-specific subgraph from knowledge bases and text for effective multi-hop reasoning in open domain question answering.

**QGG** (Lan & Jiang, 2020) introduces a method that enhances complex question answering by generating flexible query graphs for multi-hop questions and integrating constraints early.

**EmbedKGQA** (Saxena et al., 2020) introduces a method that uses knowledge graph embeddings to improve multi-hop question answering, addressing knowledge graph sparsity.

**EMQL** (Sun et al., 2020) presents a method that combines centroid-sketch entity set representations with neural retrieval over embedded knowledge base triples.

**NSM$_{+h}$** (He et al., 2021) introduces a teacher-student framework for multi-hop KBQA, where the teacher network learns intermediate supervision signals through forward and backward reasoning to enhance the student network's reasoning capability.

**GrailQA Ranking** (Gu et al., 2021) presents a BERT-based KBQA model, demonstrating the critical role of pre-trained contextual embeddings, focusing on three levels of generalization - i.i.d., compositional, and zero-shot.

**ReTraCk** (Chen et al., 2021) introduces a neural semantic parsing framework, which combines retriever, transducer, and checker components for efficient and effective KBQA.

**TransferNet** (Shi et al., 2021) introduces a model that combines a transparent, attention-based approach with the ability to handle both label and text relations in a unified framework.

**Relation Learning** (Yan et al., 2021) presents a method that integrates pre-trained language models with auxiliary tasks like relation extraction and reasoning.

**Rigel** (Sen et al., 2021) introduces a method for enhancing end-to-end question answering using differentiable knowledge graphs, and adds an intersection operation to handle multiple-entity questions more effectively.

**CBR-KBQA** (Das et al., 2021) employs a case-based reasoning framework that retrieves similar cases (questions and logical forms) from a nonparametric memory, then reuses and revises these

cases to generate logical forms for new questions, demonstrating its capability to handle complex questions and unseen relations without retraining.

**Subgraph Retrieval** (Zhang et al., 2022b) introduces a method devising a trainable subgraph retriever (SR) decoupled from the reasoning process, which efficiently retrieves relevant subgraphs for question answering, enhancing performance by focusing on more relevant and smaller subgraphs and combining with subgraph-oriented reasoners.

**RnG-KBQA** (Ye et al., 2022) introduces a framework that combines ranking and generation, using a rank-and-generate approach, where a ranker model identifies candidate logical forms and a generation model refines them.

**Program Transfer** (Cao et al., 2022) proposes a novel two-stage parsing framework with an efficient ontology-guided pruning strategy for complex KBQA, which involves a sketch parser that translates questions into high-level program sketches and an argument parser that fills in detailed arguments.

**TIARA** (Shu et al., 2022) introduces a novel method that enhances question answering over knowledge bases by using multi-grained retrieval, which improves the performance of pre-trained language models by focusing on the most relevant knowledge base contexts, including entities, logical forms, and schema items, and employs constrained decoding to control the output space, reducing generation errors and enhancing robustness in various generalization settings.

**UniK-QA** (Oguz et al., 2022) proposes a framework that integrates structured, unstructured, and semi-structured knowledge sources, such as text, tables, lists, and knowledge bases, which flattens all data into text and applies a unified retriever-reader model.

**ArcaneQA** (Gu & Su, 2022) introduces a generation-based KBQA model that addresses large search space and schema linking challenges in KBQA, which employs dynamic program induction for efficient search space navigation and dynamic contextualized encoding for improved schema linking.

**GMT-KBQA** (Hu et al., 2022b) proposes a multi-task learning framework with a shared T5 encoder to improve question answering over knowledge bases by simultaneously learning entity disambiguation, relation classification, and logical form generation.

**Uni-Parser** (Liu et al., 2022b) unifies semantic parsing for question answering on both knowledge bases and databases by using a three-module approach: primitive enumeration, ranking, and compositional generation.

**UnifiedSKG** (Xie et al., 2022) unifies 21 structured knowledge grounding tasks into a text-to-text format, leveraging T5 models and multi-task learning to improve performance across diverse tasks and facilitate zero-shot and few-shot learning investigations.

**UniKGQA** (Jiang et al., 2023b) integrates retrieval and reasoning for multi-hop question answering over knowledge graphs, employing a unified architecture that combines a semantic matching module and a matching information propagation module, enhanced by pre-training and fine-tuning strategies.

**DECAF** (Yu et al., 2023) combines the generation of logical forms and direct answers, leveraging a sequence-to-sequence framework with retrieval from linearized knowledge bases.

**BeamQA** (Atif et al., 2023) combines sequence-to-sequence prediction and beam search for multi-hop knowledge graph question answering, using a fine-tuned BART model for path generation and a novel beam search execution algorithm to traverse the knowledge graph and find answers.

**HGNet** (Chen et al., 2023) proposes a hierarchical query graph generation approach with an outlining stage for structural constraints and a filling stage for instance selection.

**SKP** (Dong et al., 2023) introduces structured knowledge-aware pre-training tasks, an efficient linearization strategy, and an interval attention mechanism, leading to significant improvements in subgraph retrieval and encoding.

**StructGPT** (Jiang et al., 2023a) enhances LLMs' reasoning over structured data using an Iterative Reading-then-Reasoning (IRR) approach, which includes specialized interfaces for efficient data access, a novel invoking-linearization-generation procedure, and iterative reasoning to effectively utilize structured data in answering complex questions.

**FC-KBQA** (Zhang et al., 2023) introduces a Fine-to-Coarse composition framework for question answering over knowledge bases, utilizing fine-grained component detection, middle-grained component constraints, and coarse-grained component composition.

**PanGu** (Gu et al., 2023) proposes a grounded language understanding framework that combines a symbolic agent and a neural language model, which allows for the incremental construction of valid plans and utilizes the language model to evaluate the plausibility of these plans.

**ToG** (Sun et al., 2023) integrates LLMs with KGs for deep and responsible reasoning, using a beam search algorithm in KG/LLM reasoning, which allows the LLM to dynamically explore multiple reasoning paths in KG and make decisions accordingly, enhancing LLMs' deep reasoning capabilities for knowledge-intensive tasks.

## B  PLUG-AND-PLAY SETTINGS

ChatKBQA has a plug-and-play characteristic, as shown in 3 parts, including the Open-source LLMs, PEFT methods, and Unsupervised Retrieval methods, all of which have different candidates. The following is a description of these candidates.

### B.1  OPEN-SOURCE LARGE LANGUAGE MODELS

In the open source macromodelling part, we choose Llama-2, ChatGLM2 and Baichuan2.

**Llama-2-7B / Llama-2-13B** (Touvron et al., 2023): Part of Meta AI's Llama series, these models are auto-regressive transformers with 7 and 13 billion parameters, trained on 2 trillion tokens. They are optimized for dialogue and general language tasks, leveraging supervised fine-tuning and reinforcement learning for better alignment with human preferences.

**ChatGLM2-6B** (Zeng et al., 2023): Developed by Tsinghua University, this 6.2 billion-parameter bilingual Chinese-English chat model improves upon its predecessor with enhanced performance, longer context support, and efficient inference. It's designed for fluent, coherent conversations in both languages.

**Baichuan2-7B / Baichuan2-13B** (Yang et al., 2023): From Baichuan Intelligent Technology, these multilingual models have 7 and 13 billion parameters and are trained on 2.6 trillion tokens. They support Chinese and English, offering competitive performance on various language processing benchmarks and are available for open-source commercial use.

### B.2  PARAMETER-EFFICIENT FINE-TUNING METHODS

In the PEFT part, we choose LoRA, QLoRA, P-tuning v2 and Freeze.

**LoRA (Low-Rank Adaptation)** (Hu et al., 2022a) is a PEFT method that introduces low-rank matrices to adapt large pre-trained models. Instead of fine-tuning all parameters, LoRA modifies only a small number of additional trainable parameters, effectively reducing the computational cost. It alters the weights of a pre-trained model in a low-rank decomposed space, allowing for efficient adaptation while maintaining the original model's structure and size.

**QLoRA (Quantized Low-Rank Adaptation)** (Dettmers et al., 2023) is an extension of LoRA, combining low-rank adaptation with quantization techniques. It aims to further reduce the computational and memory overhead associated with fine-tuning large models. By quantizing the additional low-rank matrices introduced in LoRA, QLoRA provides a more memory-efficient approach to adapting pre-trained models.

**P-tuning v2** (Liu et al., 2022a) advances the concept of prompt tuning, where trainable prompts are added to a fixed pre-trained model to guide its predictions. P-tuning v2 introduces trainable continuous prompts at the embedding layer and employs a sophisticated bi-level optimization strategy. This approach enhances the model's ability to adapt to specific tasks with minimal parameter updates, making it more efficient than traditional fine-tuning methods.

**Freeze** (Geva et al., 2021) is a parameter-efficient approach where most of the layers of a pre-trained model are frozen, and only a small fraction of the parameters are fine-tuned. This technique signif-

icantly reduces the computational resources required for fine-tuning, making it ideal for scenarios with limited computational budgets. By selectively updating only certain layers or parts of a model, Freeze retains the general knowledge of the pre-trained model while adapting it to specific tasks.

### B.3 Unsupervised Retrieval Methods

In the Unsupervised Retrieval part, we choose SimCSE, Contriever and BM25.

**SimCSE** (Gao et al., 2021) is an unsupervised method for generating sentence embeddings using contrastive learning. It enhances semantic understanding by using variations of the same sentence to train neural networks, improving performance in tasks like textual similarity and natural language inference.

**Contriever** (Izacard et al., 2022) is an unsupervised technique for creating dense passage embeddings, designed for effective retrieval in large document collections. It focuses on semantic content, offering an advanced alternative to traditional keyword-based retrieval methods.

**BM25** (Robertson & Zaragoza, 2009) is a probabilistic ranking function used in search engines. It evaluates document relevance to a search query, improving upon models like TF-IDF by incorporating document length normalization and term frequency saturation.

## C Dataset Statistics

As shown in Table 4, this is the statistical information of the two KBQA datasets, WebQSP and CWQ, made by the ChatKBQA experiment.

| Dataset | #Question | #Skeleton(LF) | #Entity | #Relation | #Train | #Valid | #Test | KB |
|---------|-----------|---------------|---------|-----------|--------|--------|-------|----|
| WebQSP | 4,737 | 34 | 2,461 | 628 | 3,098 | - | 1,639 | Freebase |
| CWQ | 34,689 | 174 | 11,422 | 845 | 27,639 | 3,519 | 3,531 | Freebase |

Table 4: Dataset statistics, where the columns respectively indicate the number of all KBQA questions, logical form skeletons, participant entities, participant relations, and questions in train/valid/test sets, followed by KB's name.

**WebQSP dataset** (Yih et al., 2016b) is developed to evaluate the importance of gathering semantic parses compared to just answers for a set of questions. WebQSP consists of 4,737 KBQA questions, with 34 logical form skeletons and 2,461 entities involved. There are 628 relations specified within the dataset, which is divided into a training set of 3,098 questions and a test set of 1,639 questions. This dataset utilizes Freebase as its knowledge base and is tailored for developing systems that can process and answer natural language questions using structured data.

**CWQ dataset** (Talmor & Berant, 2018) is designed to answer complex questions requiring reasoning over multiple web snippets, which contains a large set of complex questions in natural language and is versatile in its applications. CWQ is considerably larger with 34,689 questions, underpinned by 174 logical form skeletons. It encompasses a more extensive set of entities amounting to 11,422 and includes 845 relations. The training set comprises 27,639 questions, supplemented by a validation set of 3,519 questions and a test set of 3,531 questions. CWQ also leverages Freebase as its knowledge base and is designed for complex question answering tasks that require the interpretation and synthesis of information from various sources.

## D Hyperparameter Settings

We use the grid search method to select the optimal hyperparameter settings for the network. The F1 score of KBQA predicted without oracle entity linking is chosen as the evaluation metric. The hyperparameters that we can adjust and the possible values of the hyperparameters are first determined according to the structure of our model in Table 5.

Afterwards, the different hyperparameter choices are combined to judge the merit of the hyperparameter combinations. The optimal hyperparameter combinations of the model are obtained by

circular traversal of all hyperparameter combinations. The optimal hyperparameter combinations are shown in **bold**.

| Hyperparameter | WebQSP | CWQ |
|---|---|---|
| LLM Selection | **Llama-2-7B** | **Llama-2-13B** |
| Fine-tuning Type | {**LoRA**, QLoRA, P-tuning v2, Freeze} | {**LoRA**, QLoRA, P-tuning v2, Freeze} |
| Train Batch Size | {1, 2, 3, **4**} | {1, 2, 3, **4**} |
| Learning Rate | {**5e-5**, 5e-4, 5e-3} | {**5e-5**, 5e-4, 5e-3} |
| Train Epoch | {10, 50, **100**} | {**10**, 50, 100} |
| Test Batch Size | {**1**, 2, 3, 4} | {**1**, 2, 3, 4} |
| Beam Size | {1, 2, 5, 8, **15**} | {1, 2, 5, **8**} |
| Retrieval Type | {**SimCSE**, Contriever, BM25} | {**SimCSE**, Contriever, BM25} |
| ER Top $k_e$ | {5, 10, **50**, 100} | {5, 10, **50**, 100} |
| ER Threshold $t_e$ | {0.0, 0.0001, **0.001**, 0.01, 0.1} | {0.0, 0.0001, **0.001**, 0.01, 0.1} |
| ER Top $k_1$ | {10, 30, **50**, 100, 1000} | {10, **30**, 50, 100, 1000} |
| ER Threshold $t_1$ | {**0.0**, 0.0001, 0.001, 0.01, 0.1} | {**0.0**, 0.0001, 0.001, 0.01, 0.1} |
| RR Top $k_r$ | {3, 5, **15**, 30} | {3, 5, **15**, 30} |
| RR Threshold $t_r$ | {0.0, 0.0001, 0.001, **0.01**, 0.1} | {0.0, 0.0001, 0.001, **0.01**, 0.1} |
| RR Top $k_2$ | {30, **300**, 3000, 10000} | {40, 400, **4000**, 10000} |
| RR Threshold $k_2$ | {**0.0**, 0.0001, 0.001, 0.01, 0.1} | {**0.0**, 0.0001, 0.001, 0.01, 0.1} |

Table 5: Hyperparameter Search.

For example, WebQSP hyperparameter choices select Llama-2-7B model, as shown by bolded values, for optimal model performance. LoRA is the fine-tuning type chosen, suggesting low-rank adjustments to model parameters. A train batch size of 4, learning rate of 5e-4, and 50 training epochs indicate a preference for moderate-sized data processing batches and a faster learning rate over many epochs. Test batch size of 4 and beam size of 5 indicate evaluation and prediction generation configuration. The retrieval algorithm was SimCSE because it compares sentence embeddings well. The top-k and threshold values for Entity Retrieval (ER) and Relation Retrieval (RR) were set to balance retrieving relevant information and computational efficiency.

# E  COMPLEXITY ANALYSIS

During the retrieval phase, we measure the complexity of the algorithm using two indicators: the number of times vector similarity is calculated and the number of attempts to execute the logical form. Assuming the beam size in the generation phase is set to $b$, the size of the KB entity set is $E$, and the average logical form skeleton has $n_e$ entities, the complexity of entity retrieval is O($bn_eE$). For each entity's position, we select entities that rank in the top $k_e$ in similarity and are greater than the threshold $t_e$ for replacement. For the logical form as a whole, we select the top $k_1$ logical forms with a combined probability greater than the threshold $t_1$ as the result of entity retrieval.

In the relation retrieval phase, similarly, assuming the size of the KB relation set is R, and the average logical form skeleton has $n_r$ entities, the complexity of entity retrieval is O($k_1 n_r R$). For each position's relation, we select relations that rank in the top $k_r$ in similarity and are greater than the threshold $t_r$ for replacement. For the logical form as a whole, based on the combination probability of the relation retrieval results, we select the top $k_2$ logical forms greater than the threshold $t_2$ as the result of relation retrieval.

Therefore, the complexity of the number of vector similarity calculations is O($bn_eE + k_1 n_r R$). For the number of attempts to execute the logical form, we initially attempt with the first b logical forms; if none can be executed, we proceed with entity retrieval and attempt up to $k_1$ times. If there is still no executable logical form, we move to relation retrieval and attempt up to $k_2$ times. Thus, the complexity of the number of logical form execution attempts is O($b + k_1 + k_2$).

In this way, for KBQA tasks with large entity and relation sets, other parameters are much smaller than $E$ and $R$, making the complexity of vector similarity calculations in the order of O(n) and the complexity of logical form execution attempts in the order of O(1), both of which are controllable.

# F DISCUSSION OF LLM COMBINED WITH KG.

## F.1 INSIGHTS FROM CHATKBQA.

(1) We propose a straightforward KBQA framework that uses fine-tuned open-source large models for the first time. (2) Innovatively, we adopt a generate-then-retrieve approach to enhance generation outcomes and retrieval efficiency separately, ultimately boosting KBQA performance. (3) Our framework has plug-and-play capabilities, allowing flexible replacement of LLMs and retrieval models to address the KBQA challenge. (4) Our approach introduces a new paradigm for LLMs to conduct interpretable external knowledge-based Q&A, offering a fresh perspective on merging LLMs and KGs.

To summarize, ChatKBQA proposes a thought taking both the advantages of using LLMs to do natural language semantic parsing for graph query generation and calling external KBs to interpretably reason with queries, which we name Graph Query of Thoughts (GQoT), a promising LLM+KG combination paradigm to better utilize the external knowledge, improve Q&A's interpretability, and avoid LLM's hallucinations.

## F.2 FUTURE DIRECTIONS.

ChatKBQA still has much room for improvement, such as in the design of the training set, the decomposition of complex questions, support for various graph query languages, and applications in specific domains. These are also our future research directions:

**Training set design**: ChatKBQA is the first method to fine-tune open-source large models using unsupervised retrieval methods for the KBQA task, achieving state-of-the-art results. Therefore, the effectiveness of fine-tuning depends on the quality of the dataset used to map natural language to logical forms. In future work, we plan to enhance the training set by extracting computation graphs from the knowledge graph using graph sampling, then converting them into natural language, and exploring ways to achieve maximum training effectiveness with the least amount of training data.

**Decomposition of complex questions**: We have seen that for some simple tasks, such as one-hop and two-hop queries, ChatKBQA performs very well because the logical form skeletons involved are very similar and the fine-tuned LLM can generate them effectively. However, generating the corresponding long logical forms for more complex questions is a challenge. Therefore, in future work, we plan to use large model techniques such as CoT or Agent to decompose natural language questions into simpler logical forms for better performance.

**Support for various graph query languages**: Currently, ChatKBQA converts generated logical forms into SPARQL queries in two datasets, as the Freebase KB stores knowledge in RDF format. We will explore more KBs and datasets, such as those using the Cypher language like Neo4j, where the methodology of generating and then retrieving with ChatKBQA is also promising.

**Open-domain and specific-domain applications**: There is a demand for precision knowledge question answering in fields such as open-domain, medicine, finance, and telecommunications. We can first use UIE or LLM information extraction technology to build a knowledge graph, then fine-tune ChatKBQA to understand the structure of the knowledge graph, achieving interpretable knowledge question answering in open and specific domains.

