# OpenReview forum: "ChatKBQA: A Generate-then-Retrieve Framework for Knowledge Base Question Answering with Fine-tuned Large Language Models"
_ICLR.cc/2024/Conference — Submitted to ICLR 2024_

### Official Review · Reviewer_i3n1 · 2023-10-29

**Soundness:** 3 good
**Presentation:** 2 fair
**Contribution:** 2 fair
**Rating:** 6
**Confidence:** 3

**Summary:**

This paper proposes chatKBQA, a generate-then-retrieve KBQA framework built on fine-tuning open-source large language models (LLMs) like Llama2, chatGLM2 and Baichuan2. ChatKBQA generates logical form with fine-tuned LLMs first, then will retrieve and replace entities and relations using unsupervised retrieval. The paper shows that chatKBQA achieves SOTA on two KBQA datasets, but more details of the method should be clarified before determining whether the method truly achieves SOTA.

**Strengths:**

- ChatKBQA is very flexible, able to switch from different unsupervised retrieval methods, different language models and different efficient fine-tuning methods.
- Extensive comparison against baselines are done, and the authors provided results for different configuration on chatKBQA.

**Weaknesses:**

- In section 4.3 and 4.4, words such as “somewhat” and “good generative ability” appears in the description yet I am concerned that even with beam search, only 77% of the result lists contain the ground truth logical forms. If the relationships and entities were replaced, how do we ensure that the plugged-in entities/relationships were the right one? In what percentage were the right entities/relationships being plugged in if no ground truth is available?
- In section 4.5, the authors claim that Graph-Query-of-Thoughts are a way to improve QA’s interpretability and avoid LLM’s hallucinations, which has no evidence support in the result/analysis section. This seems to be an exaggerated claim and I am not convinced.
- Presentation of the paper needs improvement. Multiple grammatical errors, and the description of the method is confusing. Explanation of methods like QLora.etc can be moved to related work, since now it is interrupting the flow of the writing.

**Questions:**

- In section 4.2, why does chatKBQA converts the SPARQL corresponding to the test set in the KBQA dataset into logical form? Why test set?
- How do you compare against chatGPT and GPT-4? The paper says that chatGPT and GPT-4 failed to generate logical form, is this zero-shot or few-shot?
- ChatKBQA seems to rely on “ground truth logical forms”, which might be a rare resource. The datasets used are outdated since the task is now a less popular task, and I am curious of how chatKBQA will be useful in an open-domain QA era.

---

> ### Author Response · Authors · 2023-11-22
> **Response to Reviewer i3n1 (1/2)**
>
> We appreciate your suggestions and are very happy to discuss them with you. We have revised the entire paper based on your feedback and that of the other two reviewers. To address the questions you raised and prevent any misunderstandings, our clarifications are as follows:
>
> **Regarding Weakness 1 (In sections 4.3 and 4.4, words such as “somewhat” and “good generative ability” appear in the description, yet I am concerned that even with beam search, only 77% of the result lists contain the ground truth logical forms. If the relationships and entities were replaced, how do we ensure that the plugged-in entities/relationships were the right ones? In what percentage were the right entities/relationships being plugged in if no ground truth is available?):**
>
> Thank you for your suggestion. During the testing phase of ChatKBQA, our actual practice is as follows: First, we use beam search to generate b candidate logical form sets $C$, and then we try to execute these b logical forms in the order of their beam scores. The first logical form that can be executed is considered our desired logical form. If none can be executed, we proceed with entity retrieval to get a new candidate logical form set $\mathcal{C}^{\prime}$, and try execution again. If still unsuccessful, we conduct relationship retrieval to obtain $\mathcal{C}^{\prime\prime}$, and then find the answer. Therefore, with each entity and relationship retrieval, the number of test samples with executable SPARQL increases, ensuring that the insertion of entities and relationships doesn't turn a correct logical form into an incorrect one, only improving the results.
>
> Since we select logical forms to execute and obtain KBQA answers based on whether they can be converted into executable SPARQL, some entities and relationships might be correctly retrieved but may not combine correctly. Therefore, we believe that calculating the percentage of correct entities/relationships being plugged in is not as meaningful as assessing the accuracy of the final executed answers. In Section 5.3, to validate entity retrieval (ER) and relation retrieval (RR) separately, we removed ER or RR from the framework and obtained three simplified variants (ChatKBQA w/o ER, ChatKBQA w/o RR, and ChatKBQA w/o ER,RR) at four different beam sizes for comparison. As shown in Figure 4, ER and RR improve about 15 and 5 percentage points on average over no oracle entity linking in the F1 score at different beam sizes, respectively. This proves the effectiveness of ER and RR, and combined with ER, RR makes KBQA perform at its best.
>
> **Regarding Weakness 2 (In section 4.5, the authors claim that Graph-Query-of-Thoughts are a way to improve QA’s interpretability and avoid LLM’s hallucinations, which has no evidence support in the result/analysis section. This seems to be an exaggerated claim and I am not convinced.):**
>
> Thank you for pointing out this issue. We proposed Graph-Query-of-Thoughts to improve QA’s interpretability and avoid LLM’s hallucinations, intending to illustrate a new paradigm combining LLM and KG, namely using fine-tuned LLMs to generate graph database query languages, then executing them. This replaces LLM reasoning with graph database queries on the knowledge graph, aiming to improve interpretability and avoid hallucinations in QA with strict interpretability requirements. This is a concept and also the direction of our future work.
>
> To avoid misunderstandings, we have moved this content to Appendix F, Future Directions, in the updated paper.
>
> **Regarding Weakness 3 (Presentation of the paper needs improvement. Multiple grammatical errors, and the description of the method is confusing. Explanation of methods like QLora.etc can be moved to related work, since now it is interrupting the flow of the writing.):**
>
> Thank you for pointing out these issues. We have comprehensively revised the entire article, checking and correcting colloquialisms and grammatical errors based on feedback from all reviewers. Regarding explanations of methods like QLora, etc., we have moved the description of efficient fine-tuning methods in Section 4.2 to related work and Appendix B.2 to improve readability and coherence.

---

> ### Author Response · Authors · 2023-11-22
> **Response to Reviewer i3n1 (2/2)**
>
> **For Question 1 (In section 4.2, why does chatKBQA convert the SPARQL corresponding to the test set in the KBQA dataset into logical form? Why test set?):**
>
> Thank you for pointing out this issue. The term 'test set' should be changed to 'train set', and we have made this correction in the updated paper.
>
> **For Question 2 (How do you compare against ChatGPT and GPT-4? The paper says that ChatGPT and GPT-4 failed to generate logical form, is this zero-shot or few-shot?):**
>
> During the generation phase of ChatKBQA, we replaced our fine-tuned open-source LLMs with prompts to have ChatGPT and GPT-4 generate SPARQL in a zero-shot manner. As mentioned in the updated Section 5.5, ChatGPT and GPT-4, despite their large parametric capacities, struggle to generate standard logical forms effectively because they are not open-source and thus cannot be fine-tuned. They can produce SPARQL language, but constructing the correct query skeleton, entities, and relations is challenging. This difficulty arises from their limited ability to perceive the complex structure of the external knowledge base (KB) through designed prompts within a restricted context length.
>
> **For Question 3 (ChatKBQA seems to rely on “ground truth logical forms,” which might be a rare resource. The datasets used are outdated since the task is now a less popular task, and I am curious about how ChatKBQA will be useful in an open-domain QA era):**
>
> We added Appendix F to discuss the insights and future directions of ChatKBQA. ChatKBQA is the first method to fine-tune open-source LLMs using unsupervised retrieval methods for the KBQA task, achieving state-of-the-art results. Therefore, its effectiveness depends on the quality of the dataset used for mapping natural language to logical forms. In future work, we plan to enhance the training set by extracting computation graphs from the knowledge graph using graph sampling, then translating these into natural language. This approach aims to explore methods to achieve maximum training effectiveness with the least amount of training data.
>
> Furthermore, there is a growing demand for precise knowledge question-answering in various fields, including open-domain, medicine, finance, and telecommunications. We can first use technologies like UIE or LLM information extraction to build a knowledge graph. Then, by fine-tuning ChatKBQA to understand the structure of the knowledge graph, we aim to achieve interpretable knowledge question-answering in both open and specific domains.

---

> ### Author Response · Authors · 2023-11-22
> **Response to Reviewer i3n1 (Follow-up)**
>
> Thank you for your dedication and time in assessing our work and offering valuable insights!
>
> We are following up to confirm if our replies have effectively resolved your questions or concerns.
>
> Best regards, The Authors

---

### Official Review · Reviewer_wXvo · 2023-10-30

**Soundness:** 3 good
**Presentation:** 3 good
**Contribution:** 3 good
**Rating:** 6
**Confidence:** 3

**Summary:**

This paper introduces ChatKBQA, a novel generate-then-retrieve KBQA framework that leverages the power of modern fine-tuned large language models. The proposed method differs from traditional approaches in that it focuses on generating logical forms before retrieval, which overcomes inherent challenges such as retrieval inefficiencies and the misleading influence of retrieval errors in semantic parsing with the help of fine-tuned open-source LLMs and unsupervised retrieval methods. Experimental results on two standard KBQA benchmarks demonstrate that the developed framework performs better than existing methods and offers plug-and-play flexibility.

**Strengths:**

- The paper is clearly written and well organized, with sufficient background introduced before the detailed description of the methodology.
- The philosophy of generate-then-retrieve is simple and effective, significantly elevating the retrieval efficiency and reducing the retrieval error. The proposed framework achieves a new state-of-the-art performance in the KBQA domain.
- The authors conducted a detailed experimental analysis, showing the effectiveness of each designed module and the flexibility of the proposed approach as a plug-and-play framework.

**Weaknesses:**

- Lacking analysis of the failure examples, I am curious which module caused the error when the proposed framework did not get the correct answer.
- It would be better to list the detailed statistics of the datasets, *e.g.,* number of the skeletons of logical forms involved in the training set and the test set, etc.
- Please add an analysis of the computational efficiency (complexity) of the retrieval module. As far as I understand, each entity in the generated entity list has to compute similarity with the whole entity set of the knowledge base (same for the relations).

**Questions:**

- Why does the beam size have such an influence on the performance?
- During the test phase, when the fine-tuned LLMs generate logical forms, can they correctly generate relations not seen in the training set? I guess they can generate the right entities even though those entities have not appeared in the training set before, as seed entities are typically included in the natural language questions. But it is not necessarily for relations, this is why I'm curious about the result that ChatKBQA w/o RR is better than ChatKBQA w/o ER.

**Details Of Ethics Concerns:**

No ethics issues found.

---

> ### Author Response · Authors · 2023-11-22
> **Response to Reviewer wXvo**
>
> We appreciate your suggestions and are very happy to discuss them with you. We have revised the entire paper based on your feedback and that of the other two reviewers. To address the questions you raised and prevent any misunderstanding, our clarifications are as follows:
>
> **Regarding Weakness 1 (Lacking analysis of the failure examples, I am curious which module caused the error when the proposed framework did not get the correct answer.):**
>
> Thank you for your suggestion. We have added Section 5.8, expanding on the error analysis of ChatKBQA. The errors can be divided into four categories: Logical form skeleton error (40.10%), Entity retrieval error (27.17%), Relation retrieval error (19.48%), and SPARQL conversion error (13.26%).
>
> The majority of the errors are caused by ChatKBQA failing to provide the correct logical form skeleton for the problem, which is due to the limited representation of certain complex skeletons in the training set. Errors in entity and relation retrieval are due to our use of unsupervised retrieval methods. Finally, a small proportion of the remaining errors arise from the fact that, although the generated logical form is consistent with the ground truth, it fails to execute or the answers are inconsistent when converted to SPARQL, which may be caused by the loss of the conversion from logical form to SPARQL or possibly changes in the KB. Therefore, further addressing these errors is the direction of our future efforts.
>
> **Regarding Weakness 2 (It would be better to list the detailed statistics of the datasets, e.g., number of the skeletons of logical forms involved in the training set and the test set, etc.):**
>
> Thank you for your suggestion. We have added Appendix C to provide statistics on the datasets used in the experiments. This includes the number of questions, types of logical form skeletons, entities and relations involved, the scale of the training, validation, and test sets, as well as information on the databases relied upon.
>
> **Regarding Weakness 3 (Please add an analysis of the computational efficiency (complexity) of the retrieval module. As far as I understand, each entity in the generated entity list has to compute similarity with the whole entity set of the knowledge base (same for the relations).):**
>
> Thank you for your suggestion. We have added Appendix E to analyze the complexity of the retrieval part of ChatKBQA. We analyzed the number of similarity calculations and the number of times the graph database's executability is tested, which are the two most time-consuming aspects of the retrieval. According to the analysis, for KBQA tasks with large entity and relation sets, other parameters are much smaller than the size of the KB entity set E and relation set R, making the complexity of vector similarity calculations in the order of O(n) and the complexity of logical form execution attempts in the order of O(1), both of which are controllable.
>
> **For Question 1 (Why does the beam size have such an influence on the performance?):**
>
> We addressed this question in the updated Section 5.3 (ABLATION STUDY), second finding and analysis. Beam search is a heuristic algorithm usually used in sequence generation tasks, which expands the search space by generating multiple highly probable logical forms instead of only one. As shown in Figure 4, an increase in beam size enhances the likelihood of executing SPARQL queries based on candidate logical forms, improving the KBQA performance.
>
> **For Question 2 (During the test phase, when the fine-tuned LLMs generate logical forms, can they correctly generate relations not seen in the training set? I guess they can generate the right entities even though those entities have not appeared in the training set before, as seed entities are typically included in the natural language questions. But it is not necessarily for relations, this is why I'm curious about the result that ChatKBQA w/o RR is better than ChatKBQA w/o ER.):**
>
> We addressed this question in the updated Section 5.6 (ANALYSIS OF EFFICIENCY OF RETRIEVAL IN RETRIEVAL PHASE), second finding and analysis. As shown in Figure 6, RR enhances F1 score by an average of 5% across various beam sizes in ablation experiments. Although relations are rarely directly present in natural language problems, the number of thousand-level relations in the KB is still small compared to the tens of millions of entities, and the LLM perceives relational information well during fine-tuning. Thus, RR does not improve performance as much as ER, but combined with ER, RR makes KBQA perform at its best. And also, on this basis, for unseen relations, they can be obtained through the generation helped with semantic retrieval.

---

> ### Author Response · Authors · 2023-11-22
> **Response to Reviewer wXvo (Follow-up)**
>
> Thank you for your dedication and time in assessing our work and offering valuable insights!
>
> We are following up to confirm if our replies have effectively resolved your questions or concerns.
>
> Best regards, The Authors

---

### Official Review · Reviewer_EDj3 · 2023-11-01

**Soundness:** 3 good
**Presentation:** 3 good
**Contribution:** 3 good
**Rating:** 5
**Confidence:** 3

**Summary:**

The paper targets the KBQA problem. The proposed method, ChatKBQA, first generates a logical form, followed by the retrieval of entities and relations, aiming to avoid the influence of retrieval on logical form generation and to enhance retrieval efficiency. The authors fine-tune open-source LLMs using instruction tuning techniques to equip them with the capability to perceive and generate in logical form format.
The authors use an unsupervised retrieval method for entities and relations retrieval, which conducts phrase-level semantic retrieval in the entity set and relation set of the KB for entities and relations in the logical form.

**Strengths:**

1. The experiment results are stronger and better than the previous SOTA.
2. An earlier work leveraging LLM to generate a logic form for retrieval. The framework looks interesting by adjusting each component for LLM.
3. Good discussion on related work.

**Weaknesses:**

1. The proposed method is similar to semantic parsing-based methods, which focus on translating questions into logical forms executable against KBs, such as SPARQL, query graph, and S-expression, as discussed in the related work.
2. Considering the authors replaced the backbone model with LLMs, it is hard to identify which parts play the key role in performance improvement. It would be better to have more ablation studies. For example, replacing LLM with the pre-trained model in baseline, such as T5, or fixing LLM and replacing the parsing and IR component with the baseline method.
3. Efficient fine-tuning methods seem not related to the claims in the paper. There's no need to discuss it in the methodology section.

**Questions:**

As mentioned in the weaknesses, which part plays a crucial role in model improvement?

---

> ### Author Response · Authors · 2023-11-22
> **Response to Reviewer EDj3**
>
> We appreciate your suggestions and are very happy to discuss them with you. We have revised the entire paper based on your feedback and that of the other two reviewers. To address the questions you raised and prevent any misunderstanding, our clarifications are as follows:
>
> **Regarding Weakness 1 (The proposed method is similar to semantic parsing-based methods, which translate questions into logical forms executable against knowledge bases, such as SPARQL, query graph, and S-expression, as discussed in related work.):**
>
> As mentioned in the Related Work section, ChatKBQA can be categorized as an SP-based (Semantic Parsing-based) KBQA method. However, unlike previous SP-based methods, ChatKBQA introduces a new approach of generating and then retrieving to separately enhance the effects of generation and retrieval. In the generation phase, it uses a fine-tuned large language model to generate a logical form. In the retrieval phase, it employs an unsupervised method for entity and relation retrieval, achieving state-of-the-art results in the KBQA task. Additionally, ChatKBQA provides new insights into combining LLMs with KGs for interpretable knowledge question-answering.
>
> **Regarding Weakness 2 (Considering the replacement of the backbone model with LLMs, it's challenging to identify which components significantly contribute to performance improvement. More ablation studies would be beneficial, such as replacing the LLM with a pre-trained baseline model like T5, or keeping the LLM constant while substituting the parsing and IR component with the baseline method.):**
>
> We believe that ChatKBQA consists of two stages: generation and retrieval. In the updated paper, we conducted ablation analyses on four key aspects: fine-tuning of the large model, beam search, entity retrieval, and relation retrieval, as detailed in Section 5.3. The results demonstrate that each of these aspects contributes to the improvement of the KBQA performance.
>
> To further analyze the enhancements in the generation and retrieval stages of ChatKBQA, in the updated Section 5.5, we replaced the LLM component with models like ChatGPT and T5. This validates the advantages of fine-tuning open-source LLMs like Llama-2 in ChatKBQA. In Section 5.6, we changed the previous pre-retrieval to post-retrieval to verify improvements in retrieval efficiency during the retrieval phase.
>
> **Regarding Weakness 3 (Efficient fine-tuning methods do not seem relevant to the paper's claims and do not need to be discussed in the methodology section.):**
>
> Thank you for your suggestion. We have moved the content on efficient fine-tuning methods to Appendix B.2 in the latest version of the paper.
>
> **For Question 1 (As mentioned in the weaknesses, which part plays a crucial role in model improvement?):**
>
> We believe that, based on our experimental analysis, fine-tuning the large model, beam search, entity retrieval, and relation retrieval each play a significant role in enhancement. Their combined effect has enabled ChatKBQA to achieve state-of-the-art performance in the KBQA task.

---

> ### Author Response · Authors · 2023-11-22
> **Response to Reviewer EDj3  (Follow-up)**
>
> Thank you for your dedication and time in assessing our work and offering valuable insights!
>
> We are following up to confirm if our replies have effectively resolved your questions or concerns.
>
> Best regards, The Authors

---

### Author Response · Authors · 2023-11-22
**General Response**

We are grateful to the three reviewers for dedicating their valuable time to provide guidance and advice. We greatly appreciate the recognition of our work's novelty in motivation, state-of-the-art (SOTA) results, flexible framework, and extensive experimental verification by the reviewers.

**Our main contributions are summarized as follows:**

1.We introduce a straightforward Knowledge-Based Question Answering (KBQA) framework that, for the first time, employs fine-tuned open-source large models.

2.We use a generate-then-retrieve approach to separately enhance generation outcomes and retrieval efficiency, ultimately boosting KBQA performance.

3.Our framework is designed with plug-and-play capabilities, allowing the flexible replacement of Large Language Models (LLMs) and retrieval models to effectively tackle the KBQA challenge.

4.Our approach pioneers a new paradigm for LLMs to conduct interpretable external knowledge-based Q&A, offering a fresh perspective on integrating LLMs and Knowledge Graphs (KGs).

**To further and comprehensively improve the quality of our paper, we have incorporated valuable suggestions from all three reviewers and made the following modifications:**

1.We have moved the description of the efficient fine-tuning method in Section 4.2 and the description of the unsupervised retrieval method in Section 4.3 to **Appendices B.2 and B.3**, respectively, to improve the readability and rationality of these sections.

2.In the Experiment section, we have conducted further analysis on the experiments in **Sections 5.3, 5.4, 5.5, and 5.6**. The experimental findings are now more prominently displayed with bolded titles to clearly and accurately reflect the role of each ChatKBQA module in improving results and their comparison with other methods in generation and retrieval stages.

3.We added **Section 5.8** for an error analysis of ChatKBQA, indicating the stage at which each error occurs.

4.We included **Appendix C** to present detailed statistical information about the dataset, logical form skeletons, and more.

5.**Appendix D** has been added to demonstrate the parameter selection for ChatKBQA.

6.We added **Appendix E** to analyze the complexity of similarity calculations and SparQL execution during the retrieval stage of ChatKBQA, demonstrating that retrieval complexity is controllable.

7.**Appendix F** has been included to summarize the new insights ChatKBQA provides for the LLM+KG approach, outline current limitations in fine-tuning LLMs for KBQA, and suggest future directions for development.

8.We have thoroughly revised the entire paper, addressing colloquialisms and grammatical errors based on the reviewers' feedback.

---

### Meta-Review · Area_Chair_jJqs · 2023-12-06

**Metareview:**

The paper proposes a method for question answering from knowledge bases (KB-QA) based on semantic parsing. The initial semantic parses are generated by a fine-tuned LLM, followed by an unsupervised entity and relation retrieval phase. Results show incremental improvements over previous KG-QA results. The main weaknesses are limited novelty in the methodology and that the paper's experiments do not show clearly enough that the gains are coming from the proposed methods.

**Justification For Why Not Higher Score:**

The paper's experiments do not show clearly enough that the gains are coming from the proposed methods.

**Justification For Why Not Lower Score:**

N/A

---

### Decision · Program_Chairs · 2024-01-16

Reject